# Characterization of the transcriptome, nucleotide sequence polymorphism, and natural selection in the desert adapted mouse *Peromyscus eremicus*

Matthew D. MacManes[1] and Michael B. Eisen[2]

[1] Department of Molecular, Cellular and Biomedical Sciences, University of New Hampshire, Durham, NH, USA

[2] Howard Hughes Medical Institute, University of California, Berkeley, CA, USA

## ABSTRACT

As a direct result of intense heat and aridity, deserts are thought to be among the most harsh of environments, particularly for their mammalian inhabitants. Given that osmoregulation can be challenging for these animals, with failure resulting in death, strong selection should be observed on genes related to the maintenance of water and solute balance. One such animal, *Peromyscus eremicus*, is native to the desert regions of the southwest United States and may live its entire life without oral fluid intake. As a first step toward understanding the genetics that underlie this phenotype, we present a characterization of the *P. eremicus* transcriptome. We assay four tissues (kidney, liver, brain, testes) from a single individual and supplement this with population level renal transcriptome sequencing from 15 additional animals. We identified a set of transcripts undergoing both purifying and balancing selection based on estimates of Tajima's D. In addition, we used the branch-site test to identify a transcript—Slc2a9, likely related to desert osmoregulation—undergoing enhanced selection in *P. eremicus* relative to a set of related non-desert rodents.

## INTRODUCTION

Deserts are widely considered one of the harshest environments on Earth. Animals living in desert environments are forced to endure intense heat and drought, and as a result, species living in these environments are likely to possess specialized mechanisms to deal with them. While living in deserts likely involves a large number of adaptive traits, the ability to osmoregulate—to maintain the proper water and electrolyte balance—appears to be paramount (*Walsberg, 2000*). Indeed, the maintenance of water balance is one of the most important physiologic processes for all organisms, whether they be desert inhabitants or not. Most animals are exquisitely sensitive to changes in osmolality, with slight derangement eliciting physiologic compromise. When the loss of water exceeds dietary intake, dehydration—and in extreme cases, death—can occur. Thus there has likely been strong selection for mechanisms supporting optimal osmoregulation in species that

Corresponding author
Matthew D. MacManes,
macmanes@gmail.com, Twitter:
@PeroMHC

live where water is limited. Understanding these mechanisms will significantly enhance our understanding of the physiologic processes underlying osmoregulation in extreme environments, which will have implications for studies of human health, conservation, and climate change.

The genes and structures responsible for the maintenance of water and electrolyte balance are well characterized in model organisms such as mice (*Tatum et al., 2009*), rats (*Romero et al., 2007*; *Rojek et al., 2006*; *Nielsen et al., 1995*), and humans (*Mobasheri et al., 2007*; *Bedford, Leader & Walker, 2003*; *Nielsen et al., 1999*). These studies, many of which have been enabled by newer sequencing technologies, provide a foundation for studies of renal genomics in non-model organisms. Because researchers have long been interested in desert adaptation, a number of studies have looked at the morphology or expression of single genes in the renal tissues of desert adapted rodents *Phyllotis darwini* (*Gallardo, Cortés & Bozinovic, 2005*), *Psammomys obesus* (*Kaissling et al., 1975*), and *Perognathus penicillatus* (*Altschuler et al., 1979*). More recently, full renal transcriptomes have been generated for *Dipodomys spectabilis* and *Chaetodipus baileyi*, (*Marra, Romero & DeWoody, 2014*) as well as *Abrothrix olivacea* (*Giorello et al., 2014*).

These studies provide a rich context for current and future work aimed at developing a synthetic understanding of the genetic and genomic underpinnings of desert adaptation in rodents. As a first step, we have sequenced, assembled, and characterized the transcriptome (using four tissue types—liver, kidney, testes and brain) of a desert adapted cricetid rodent endemic to the southwest United States, *Peromyscus eremicus*. These animals have a lifespan typical of small mammals (*Veal & Caire, 2001*), and therefore an individual may live its entire life without ever drinking water. Additionally, they have a distinct advantage over other desert animals (e.g., *Dipodomys*) in that they breed readily in captivity, which enables future laboratory studies of the phenotype of interest. In addition, the focal species is positioned in a clade of well known animals (e.g., *P. californicus*, *P. maniculatus*, and *P. polionotus*) (*Feng et al., 2007*) with growing genetic and genomic resources (*Shorter et al., 2014*; *Panhuis et al., 2011*; *Shorter et al., 2012*). Together, this suggests that future comparative studies are possible.

While the elucidation of the mechanisms underlying adaptation to desert survival is beyond the scope of this manuscript, we aim to lay the groundwork by characterizing the transcriptome from four distinct tissues (brain, liver, kidney, testes). These data will be included in the current larger effort aimed at sequencing the entire genome. Further, via sequencing the renal tissue of a total of 15 additional animals, we characterize nucleotide polymorphism and genome-wide patterns of natural selection. Together, these investigations will aid in our overarching goal to understand the genetic basis of adaptation to deserts in *P. eremicus*.

## MATERIALS AND METHODS

### Animal collection and study design

To begin to understand how genes may underlie desert adaptation, we collected 16 adult individuals (9 male, 7 female) from a single population of *P. eremicus* over a two-year time

period (2012–2013). These individuals were captured in live traps and then euthanized using isoflurane overdose and decapitation. Immediately post-mortem, the abdominal and pelvic organs were removed, cut in half (in the case of the kidneys), placed in RNAlater and flash frozen in liquid nitrogen. Removal of the brain, with similar preservation techniques, followed. Time from euthanasia to removal of all organs never exceeded five minutes. Samples were transferred to a −80C freezer at a later date. These procedures were approved by the Animal Care and Use Committee located at the University of California Berkeley (protocol number R224) and University of New Hampshire (protocol number 130902) as well as the California Department of Fish and Game (protocol SC-008135) and followed guidelines established by the American Society of Mammalogy for the use of wild animals in research (*Sikes et al., 2011*).

## RNA extraction and sequencing

Total RNA was extracted from each tissue using a TRIzol extraction (Invitrogen) following the manufacturer's instructions. Because preparation of an RNA library suitable for sequencing is dependent on having high quality, intact RNA, a small aliquot of each total RNA extract was analyzed on a Bioanalyzer 2100 (Agilent, Palo Alto, CA, USA). Following confirmation of sample quality, the reference sequencing libraries were made using the TruSeq stranded RNA prep kit (Illumina), while an unstranded TruSeq kit was used to construct the other sequencing libraries. A unique index was ligated to each sample to allow for multiplexed sequencing. Reference libraries ($n = 4$ tissue types from Peer360, a male mouse used for generating a genome sequence—not part of the current study) were then pooled to contain equimolar quantities of each individual library and submitted for Illumina sequencing using two lanes of 150nt paired end sequencing employing the rapid-mode of the HiSeq 2500 sequencer at The Hubbard Center for Genome Sciences (University of New Hampshire). The remaining 15 libraries were multiplexed and sequenced in a mixture of 100nt paired and single end sequencing runs across several lanes of an Illumina HiSeq 2000 at the Vincent G. Coates Genome Center (University of California, Berkeley).

## Sequence data preprocessing and assembly

The raw sequence reads corresponding to the four tissue types were error corrected using the software bless version 0.17 (*Heo et al., 2014*) using kmer = 25, based on the developer's default recommendations (https://github.com/macmanes/pero_transcriptome/blob/master/analyses.md#error-correction). The error-corrected sequence reads were adapter and quality trimmed following recommendations from *MacManes (2014)* and *Mbandi et al. (2014)*. Specifically, adapter sequence contamination and low quality nucleotides (defined as Phred <2) were removed using the program Trimmomatic version 0.32 (*Bolger, Lohse & Usadel, 2014*). Reads from each tissue were assembled using the Trinity version released 17 July 2014 (*Haas et al., 2013*). We used flags to indicate the stranded nature of sequencing reads and set the maximum allowable physical distance between read pairs to 999nt (https://github.com/macmanes/pero_transcriptome/blob/master/analyses.md#trinity-assemblies). We elected to assemble reads derived from a single
deeply sequenced individual (Peer360, a male) to reduce polymorphism and thus the complexity of the de Bruijn graph, which has important implications for runtime, hardware requirements (*Lowe, Swalla & Brown, 2014*; *Pop, 2009*), and assembly contiguity (*Vijay et al., 2013*). Individual tissues were assembled independently, as we hypothesize that tissue specific isoforms would be reconstructed with higher fidelity than if all tissues were assembled together.

The assembly was conducted on a linux workstation with 64 cores and 512Gb RAM. To filter the raw sequence assembly, we downloaded *Mus musculus* cDNA and ncRNA datasets from Ensembl (ftp://ftp.ensembl.org/pub/release-75/fasta/mus_musculus/) and the *Peromyscus maniculatus* reference transcriptome from NCBI (ftp://ftp.ncbi.nlm. nih.gov/genomes/Peromyscus_maniculatus_bairdii/RNA/). We used a blastN (version 2.2.29+) procedure (default settings, evalue set to $10^{-10}$) to identify contigs in the *P. eremicus* dataset likely to be biological in origin (https://github.com/macmanes/pero_ transcriptome/blob/master/analyses.md#blasting). This procedure, when a reference dataset is available, retains more putative transcripts than a strategy employing expression-based filtering (remove if transcripts per million (TPM) <1 (*MacManes & Lacey, 2012*)) of the raw assembly. We then concatenated the filtered assemblies from each tissue into a single file and reduced redundancy using the software cd-hit-est version 4.6 (*Li & Godzik, 2006*) using default settings, except that sequences were clustered based on 95% sequence similarity (https://github.com/macmanes/pero_transcriptome/blob/master/analyses.md# cd-hit-est). This multi-fasta file was used for all subsequent analyses, including annotation and mapping.

## Assembled sequence annotation

The filtered assemblies were annotated using the default settings of the blastN algorithm (*Camacho et al., 2009*) against the Ensembl cDNA and ncRNA datasets described above, downloaded on 1 August 2014. Among other things, the Ensemble transcript identifiers were used in the analysis of gene ontology conducted in the PANTHER package (*Mi, 2004*). Next, because rapidly evolving nucleotide sequences may evade detection by blast algorithms, we used HMMER3 version 3.1b1 (*Wheeler & Eddy, 2013*) to search for conserved protein domains contained in the dataset using the Pfam database (*Punta et al., 2012*) (https://github.com/macmanes/pero_transcriptome/blob/master/analyses.md# hmmer3pfam). Lastly, we extracted putative coding sequences using Transdecoder version 4 Jul 2014 (http://transdecoder.sourceforge.net/) (https://github.com/macmanes/pero_ transcriptome/blob/master/analyses.md#transdecoder).

To identify patterns of gene expression unique to each tissue type, we mapped sequence reads from each tissue type to the reference assembly using bwa-mem (version cloned from Github 7/1/2014) (*Li, 2013*). We estimated expression for the four tissues individually using default settings of the software eXpress version 1.51 (*Roberts & Pachter, 2013*). Interesting patterns of expression, including instances where expression was limited to a single tissue type, were identified and visualized.

## Population genomics

In addition to the reference individual sequenced at four different tissue types, we sequenced 15 other conspecific individuals from the same population in Palm Desert, California. Sequence data were mapped to the reference transcriptome using bwa-mem. The alignments were sorted and converted to BAM format using the samtools software package (*Li et al., 2009*), then passed to the program ANGSD version 0.610, which was used for calculating the folded site frequency spectrum (SFS) and Tajima's D (*Korneliussen et al., 2013*) using instructions found at http://popgen.dk/angsd/index.php/Tajima.

## Natural selection

To characterize natural selection on several genes related to water and ion homeostasis, we identified several of the transcripts identified as experiencing positive selection in a recent work on desert-adapted Heteromyid rodents (*Marra, Romero & DeWoody, 2014*). The coding sequences corresponding to these genes, Solute Carrier family 2 member 9 (Slc2a9), the Vitamin D3 receptor (Vdr) and several of the Aquaporin genes (Aqp1,2,4,9), were extracted from the dataset, aligned using the software MACSE version 1.01b (*Ranwez et al., 2011*) to homologous sequences in *Mus musculus*, *Rattus norvegicus*, *Peromyscus maniculatus*, and *Homo sapiens* as identified by the conditional reciprocal best blast procedure (CRBB (*Aubry et al., 2014*)). An unrooted gene tree with branch lengths was constructed using the online resource ClustalW2-Phylogeny (http://www.ebi.ac.uk/Tools/phylogeny/clustalw2_phylogeny/), and the tree and alignment were analyzed using the branch-site model (model = 2, nsSites = 2, fix_omega = 0 versus model = 2, nsSites = 2, fix_omega = 1, omega = 1) implemented in PAML version 4.8 (*Yang & dos Reis, 2011*; *Yang, 2007*). Significance was evaluated via the use of the likelihood ratio test.

# RESULTS AND DISCUSSION

## RNA extraction, sequencing, assembly, mapping

RNA was extracted from the hypothalamus, renal medulla, testes, and liver from each individual using sterile technique. TRIzol extraction resulted in a large amount of high quality (RIN $\geq$ 8) total RNA, which was then used as input. Libraries were constructed as per the standard Illumina protocol and sequenced as described above. The number of reads per library varied from 56 million strand-specific paired-end reads in Peer360 kidney, to 9 million single-end reads in Peer321 (Table 1). Adapter sequence contamination and low-quality nucleotides were eliminated which resulted in a loss of <2% of the total number of reads. These trimmed reads served as input for all downstream analyses.

Transcriptome assemblies for each tissue type were accomplished using the program Trinity (*Haas et al., 2013*). The raw assemblies for brain, liver, testes, and kidney contained 185,425, 222,096, 180,233, and 514,091 assembled sequences respectively. This assembly was filtered using a blastN procedure against the *Mus* cDNA and ncRNA and *P. maniculatus* cDNAs, which resulted in a final dataset containing 68,331 brain-derived transcripts, 71,041 liver-derived transcripts, 67,340 testes-derived transcripts, and 113,050 kidney-derived transcripts. Mapping the error-corrected adapter/quality trimmed

**Table 1** The number of sequencing reads per sample, whose identity is indicated by Peer[number].

| DATASET | NUM. RAW READS | SRA ACCESSION |
|---|---|---|
| PEER360 TESTES | 32M PE/SS | SRR1575398 |
| PEER360 LIVER | 53M PE/SS | SRR1575397 |
| PEER360 KIDNEY | 56M PE/SS | SRR1575396 |
| PEER360 BRAIN | 23M PE/SS | SRR1575395 |
| PEER305 | 19M PE | SRR1575434 |
| PEER308 | 15M PE | SRR1575437 |
| PEER319 | 14M PE | SRR1575439 |
| PEER321 | 9M SE | SRR1575441 |
| PEER340 | 16M PE | SRR1575443 |
| PEER352 | 14M PE | SRR1575464 |
| PEER354 | 9M SE | SRR1575466 |
| PEER359 | 14M PE | SRR1575492 |
| PEER365 | 16M PE | SRR1575493 |
| PEER366 | 16M PE | SRR1575494 |
| PEER368 | 14M PE | SRR1575624 |
| PEER369 | 14M PE | SRR1575625 |
| PEER372 | 17M SE | SRR1576070 |
| PEER373 | 23M SE | SRR1576071 |
| PEER380 | 16M SE | SRR1576072 |
| PEER382 | 14M SE | SRR1576073 |

**Notes.**
PE, paired end; SS, strand specific; SE, single end sequencing.

reads to these datasets resulted in mapping 94.98% (87.01% properly paired) of the brain-derived reads to the brain transcriptome, 96.07% (88.13% properly paired) of the liver-derived reads to the liver transcriptome, 96.81% (85.10% properly paired) of the testes-derived reads to the testes transcriptome, and 91.87% (83.77% properly paired) of the kidney-derived reads to the kidney transcriptome. Together, these statistics suggest that the tissue-specific transcriptomes are of extremely high quality. All tissue-specific assemblies are available on Dryad (10.5061/dryad.qf1dp).

We then estimated gene expression on each of these tissue-specific datasets, which allowed us to understand expression patterns in the multiple tissues (Pero.tissue.xprs, available on Dryad (10.5061/dryad.qf1dp)). Specifically, we constructed a Venn diagram (Fig. 1) that allowed us to visualize the proportion of genes whose expression was limited to a single tissue and those whose expression was ubiquitous. 66,324 transcripts are expressed in all tissue types, while 13,086 are uniquely expressed in the kidney, 2,222 in the testes, 3,580 in the brain, and 2,798 in the liver. The kidney appears to an outlier in the number of unique sequences, though this could be the result of the recovery of more lowly expressed transcripts or isoforms.

In addition to this, we estimated mean TPM (number of transcripts per million) for all transcripts. Table 2 consists of the 10 genes whose mean TPM was the highest. Several genes in this list are predominately present in a single tissue type. For instance

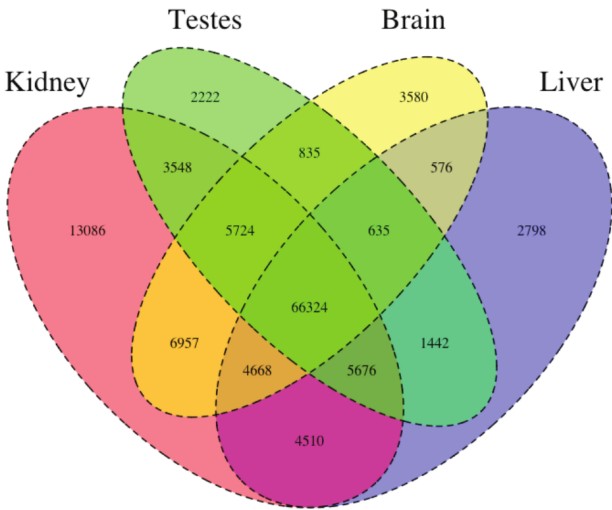

**Figure 1 The Venn Diagram, which provides a visual representation of the overlap of expression of the four tissue types.** The majority of transcripts (66,324) are expressed in all studied tissue types.

**Table 2 The 10 transcripts with the highest mean TPM (transcripts per million).**

| Transcript ID | Testes | Liver | Kidney | Brain | Genbank ID | Gene ID |
|---|---|---|---|---|---|---|
| Transcript_83842 | 2.05E+03 | 6.40E+03 | 1.03E+04 | 5.47E+03 | DQ073446.1 | COX2 |
| Transcript_126459 | 1.43E+01 | 2.22E+04 | 2.77E+01 | 6.73E+00 | XM_006991665.1 | Alb |
| Transcript_128937 | 4.39E+00 | 1.91E+04 | 4.74E+02 | 2.23E+00 | XM_007627625.1 | Apoa2 |
| Transcript_81233 | 1.71E+03 | 5.23E+03 | 6.11E+03 | 3.08E+03 | XM_006993867.1 | Fth1 |
| Transcript_94125 | 3.67E+01 | 1.08E+04 | 2.09E+03 | 2.75E+00 | XM_006977178.1 | CytP450 |
| Transcript_119945 | 5.03E+03 | 1.15E+03 | 1.33E+03 | 3.71E+03 | XM_008686011.1 | Ubb |
| Transcript_5977 | 4.95E+00 | 1.01E+04 | 3.05E+02 | 3.58E+02 | XM_006978668.1 | Tf |
| Transcript_4057 | 2.62E+01 | 9.32E+03 | 1.34E+02 | 8.38E+01 | XM_006994871.1 | Apoc1 |
| Transcript_112523 | 4.07E+02 | 7.36E+03 | 7.78E+02 | 9.54E+02 | XM_006994872.1 | Apoe |
| Transcript_98376 | 1.98E+00 | 8.66E+03 | 1.02E+00 | 2.68E+00 | XM_006970208.1 | Ttr |

Transcript_126459, Albumin is very highly expressed in the liver, but less so in the other tissues. It should be noted, however, that making inference based on uncorrected values for TPM is not warranted. Statistical testing for differential expression was not implemented due to the fact that no replicates are available.

After expression estimation, the filtered assemblies were concatenated together, and after the removal of redundancy with cd-hit-est, 122,584 putative transcripts remained available on Dryad (10.5061/dryad.qf1dp). From this filtered concatenated dataset, we extracted 71,626 putative coding sequences (72Mb, Dryad: 10.5061/dryad.qf1dp). Of these 71,626 sequences, 38,221 contained complete open reading frames (containing both start and stop codons), while the others were either truncated at the 5-prime end (20,239 sequences), the 3-prime end (6,445 sequences), or were internal (6,721 sequencing with neither stop nor start codon). The results of a Pfam search conducted on the predicted amino acid sequences are available on Dryad (10.5061/dryad.qf1dp).

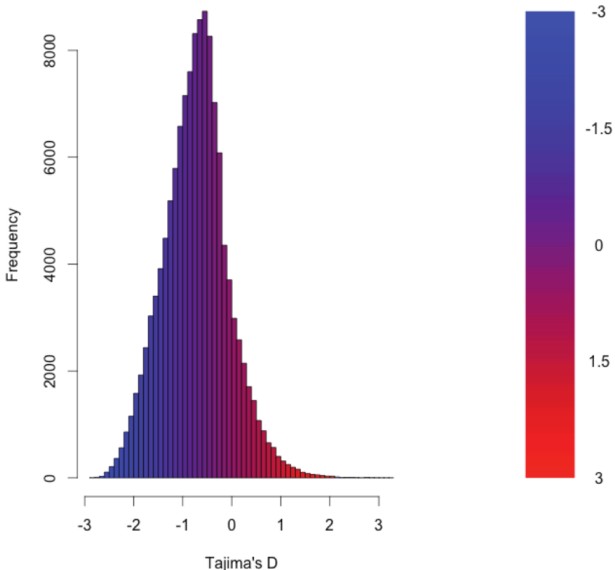

**Figure 2** **The distribution of Tajima's D for all putative transcripts.**

## Population genomics

As detailed above, RNAseq data from 15 individuals were mapped to the reference transcriptome with the resulting BAM files being used as input to the software package ANGSD. The Tajima's D statistic was calculated for all transcripts covered by at least 14 of the 15 individuals. In brief, a negative Tajima's D—a result of lower than expected average heterozygosity—is often associated with purifying or directional selection, recent selective sweep or recent population expansion, or a complex combination of these forces. In contrast, a positive value for Tajima's D represents higher than expected average heterozygosity, often associated with balancing selection.

The distribution of the estimates of Tajima's D for all of the assembled transcripts is shown in Fig. 2. Although Tajima's D is known to be sensitive to demographic history, which is largely unknown for this population, the estimates may also be drive by patterns of selection. In general, the distribution is skewed toward negative values (mean = −0.89, variance = 0.58), which may be the result of purifying selection, a model of evolution commonly invoked for coding DNA sequences (*Chamary, Parmley & Hurst, 2006*). Table 3 presents the 10 transcripts whose estimate of Tajima's D is the greatest, while Table 4 presents the 10 transcripts whose estimate of Tajima's D is the least. The former list of genes may contain transcripts experiencing balancing selection in the studied population. This list includes, interestingly, genes obviously related to solute and water balance (e.g., Clcnkb and a transmembrane protein gene) and immune function (a interferon-inducible GTPase and a Class 1 MHC gene). The latter group, containing transcripts whose estimates of Tajima's D are the smallest are likely experiencing purifying selection. Many of these transcripts are involved in core regulatory functions where mutation may have strongly negative fitness consequences.

**Table 3  The 10 transcripts with the highest values for Tajima's D, which suggests balancing selection.**

| Transcript ID | GenBank ID | Description | Tajima's D |
|---|---|---|---|
| Transcript_49049 | XM_006533884.1 | heterogeneous nuclear ribonucleoprotein H1 (Hnrnph1) | 3.26 |
| Transcript_38378 | XM_006522973.1 | Son DNA binding protein (Son) | 3.19 |
| Transcript_126187 | NM_133739.2 | transmembrane protein 123 (Tmem123) | 3.02 |
| Transcript_70953 | XM_006539066.1 | chloride channel Kb (Clcnkb) | 2.96 |
| Transcript_37736 | XM_006997718.1 | h-2 class I histocompatibility antigen | 2.92 |
| Transcript_21448 | XM_006986148.1 | zinc finger protein 624-like | 2.84 |
| Transcript_47450 | NM_009560.2 | zinc finger protein 60 (Zfp60) | 2.82 |
| Transcript_122250 | XM_006539068.1 | chloride channel Kb (Clcnkb) | 2.81 |
| Transcript_78367 | XM_006496814.1 | CDC42 binding protein kinase alpha (Cdc42bpa) | 2.78 |
| Transcript_96470 | XM_006987129.1 | interferon-inducible GTPase 1-like | 2.77 |

**Table 4  The 10 transcripts with the lowest values for Tajima's D, which suggests purifying or directional selection.**

| Transcript ID | GenBank ID | Description | Tajima's D |
|---|---|---|---|
| Transcript_84359 | XM_006991127.1 | nuclear receptor coactivator 3 (Ncoa3) | −2.82 |
| Transcript_87121 | XM_006970128.1 | methyl-CpG binding domain protein 2 (Mbd2) | −2.82 |
| Transcript_125755 | EU053203.1 | alpha globin gene cluster | −2.78 |
| Transcript_87128 | XM_006976644.1 | membrane-associated ring finger (March5) | −2.76 |
| Transcript_55468 | XM_006978377.1 | Vpr binding protein (Vprbp) | −2.75 |
| Transcript_116042 | XM_006980811.1 | membrane associated guanylate kinase (Magi3) | −2.75 |
| Transcript_18966 | XM_006982814.1 | ubiquitin protein ligase E3 component n-recognin 5 (Ubr5) | −2.75 |
| Transcript_122204 | XM_008772511.1 | zinc finger protein 612 (Zfp612) | −2.75 |
| Transcript_100550 | XM_006971297.1 | bromodomain adjacent to zinc finger domain, 1B (Baz1b) | −2.74 |
| Transcript_33267 | XM_006975561.1 | pumilio RNA-binding family member 1 (Pum1) | −2.75 |

## Natural selection

To begin to test the hypothesis that selection on transcripts related to osmoregulation is enhanced in the desert adapted *P. eremicus*, we calculated Tajima's D as described above, and implemented the branch-site test using alignments produced in MACSE. These alignments were manually inspected, and were relatively free from indels and internal stop codons. We set the sequence corresponding to *P. eremicus* for Slc2a9, Vdr, and several of the Aquaporin genes (Aqp1,2,4,9) as the foreground lineages in six distinct program executions. The transcripts Slc2a9 and Vdr were chosen specifically because they—the former significantly—were recently linked to osmoregulation in a desert rodent (*Marra, Romero & DeWoody, 2014*). The test for Slc2a9 was highly significant ($2\Delta Lnl = 51.4$, $df = 1$, $p = 0$, Table 5), indicating enhanced selection in *P. eremicus* relative to the other

**Table 5** Several candidate genes were evaluated using Tajima's D and the branch site test implemented in PAML.

| Transcript ID | Description | Tajima's D | Branch site test $p$-value |
|---|---|---|---|
| Transcript_106085 | Slc2a9 | 2.15 | $p = 0$ |
| Transcript_114624 | Vdr | 1.97 | $p = 1$ |
| Transcript_128972 | Aqp1 | 1.39 | $p = 1$ |
| Transcript_33960 | Aqp2 | 1.78 | $p = 1$ |
| Transcript_22154 | Aqp4 | 2.10 | $p = 1$ |
| Transcript_107677 | Aqp9 | 2.06 | $p = 1$ |

lineages. The branch site tests for positive selection conducted on the Vdr and Aquaporin genes were non-significant. While the branch site test of positive selection is largely non-significant, estimating Tajima's D for these few candidate loci demonstrates that either a selective or demographic process may be influencing the genome at these functionally relevant sites.

## CONCLUSIONS

As a direct result of intense heat and aridity, deserts are thought to be amongst the harshest environments, particularly for mammalian inhabitants. Given that osmoregulation can be challenging for these animals—with failure resulting in death—strong selection should be observed on genes related to the maintenance of water and solute balance. This study aimed to characterize the transcriptome of a desert-adapted rodent species, *P. eremicus*. Specifically, we characterized the transcriptome of four tissue types (liver, kidney, brain, and testes) from a single individual and supplemented this with population-level renal transcriptome sequencing from 15 additional animals. We identified a set of transcripts undergoing both purifying and balancing selection based on Tajima's D. In addition, we used a branch site test to identify a transcript, likely related to desert osmoregulation, undergoing enhanced selection in *P. eremicus* relative to a set of non-desert rodents.

## ACKNOWLEDGEMENTS

This manuscript was greatly improved by careful review from C Titus Brown, Elijah Lowe, and an anonymous reviewer, as well as by Matthew Hahn, who provided feedback on an earlier version of the manuscript posted on bioRxiv.

### Funding

MDM was supported by a NIH NRSA postdoctoral fellowship (5 F32 DK093227-03) and by startup funds provided by the University of New Hampshire. MBE is a Howard Hughes Medical Institute investigator. The funders had no role in study design, data collection and analysis, decision to publish, or preparation of the manuscript.

## Grant Disclosures

The following grant information was disclosed by the authors:
NIH NRSA postdoctoral fellowship: 5 F32 DK093227-03.

## Competing Interests

The authors declare there are no competing interests.

## Author Contributions

- Matthew D. MacManes conceived and designed the experiments, performed the experiments, analyzed the data, contributed reagents/materials/analysis tools, wrote the paper, prepared figures and/or tables, reviewed drafts of the paper.
- Michael B. Eisen contributed reagents/materials/analysis tools, reviewed drafts of the paper.

## Animal Ethics

The following information was supplied relating to ethical approvals (i.e., approving body and any reference numbers):

The work was approved by the University of New Hampshire IACUC under protocol number 130902 and University of California IACUC protocol number R224.

## Field Study Permissions

The following information was supplied relating to field study approvals (i.e., approving body and any reference numbers):

The work was approved by the California Department of Fish and Game protocol number SC-008135.

## DNA Deposition

The following information was supplied regarding the deposition of DNA sequences:

NCBI SRA BioProject: PRJNA242486.

## Data Deposition

The following information was supplied regarding the deposition of related data:

Dryad repo: http://dx.doi.org/10.5061/dryad.qf1dp.

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
