# Peer review of "Characterization of the transcriptome, nucleotide sequence polymorphism, and natural selection in the desert adapted mouse Peromyscus eremicus"

_PeerJ, doi:10.7717/peerj.642_

## Round 0.1 · original submission · Minor Revisions

Please address the various points made by the reviewers. While they are numerous, none seem particularly major, so I consider this a minor revision. Please pay special attention to the comment of reviewer 1 about having the various parts of the analysis in a repository. PeerJ generally expects authors to make all data and analysis methods available to all readers.

·

Basic reporting

The authors aim to identify genetic mechanisms behind osmoregulation using a species of dessert rodents. This species was chosen because of their adaption to extreme water deprivation, even to the point of living their lives without water. This makes a good model to identify genes under selection for osmoregulation. Studies have been done previously in model organism such as mouse, rat and human, but only on a gene-by-gene basis. This manuscript looks at transcriptome-wide differential expression, selection and nucleotide polymorphism using next-generation sequencing in efforts to advance this area of research.

1. The authors do a good job at framing their work, showing why the study is needed, the limitations and the how the work will/can lead to future research.
2. The assembly and annotation steps were well thought out. Assemblies were error corrected, quality filtered and several steps were implemented for annotation using closely related species, Pfam database and extraction of putative coding sequences. The only thing I wonder is why didn’t the authors pool the samples when assembling. This would not change their downstream pipeline much, however, it would help to recover low expressed transcripts. (Are there any citations for this?) Also, I do not understand if or why the addition reads for kidney were not used for assembly.
3. The author mentioned in results line 185 “The kidney appears to [be] an outlier in the number of unique sequences, though this could […] result [from] the recovery of more lowly expressed transcripts [caused by] deeper sequencing.” Why would this not also be the case for liver, which only has 3M (5%) less sequences?
4. I am trying to understand the filtering process for the assembled reads. From my understanding (Page 4, lines 103:109) sequences were filtered using Blastn, (Page 4, lines 113:120) annotated using Blastn, HMMER3 and Transdecoder. Is my understanding correct? If so, why were the assembled sequences filtered with Blastn before annotated with Blastn and HMMER3? I thought the point of HMMER3 was to retain divergent sequences not detected by blastn.
5. For the natural section results, I think it would be interest to add more than two genes. Perhaps the top and bottom 10 genes from the Tajima’s D analysis.
6. It would also be nice to have the various parts of the analysis in a repository, for reviewing and open science purposes.

Overall I believe it is a good paper with interesting analysis, and cool results.


Elijah K Lowe and C. Titus Brown

Experimental design

Some more explicit details to enable replication would be welcome, as described above.

Validity of the findings

No comments

Additional comments

No comments.

Reviewer 2 ·

Basic reporting

Major comments in basic reporting section:

1. Citation format should match the "name, year" format described for the journal, currently it is in a different, numbered format

2. Introduction, lines 46-47: In discussing that P. eremicus does not drink water, is there a study or citation that gives their lifespan and/or drinking habits? Are the authors referring back to the species account cited in the previous section?

3. In reporting the individuals captured the authors should provide some metadata such as age (juvenile vs. adult) and sex (were there equal numbers of each sex, or more of one sex than the other)?

4. In the methods lines 86-89, the specific multiplexing and number of lanes of sequencing should be reported (how many individuals were sequenced on each lane, etc.) Perhaps this information could be included in table 1.

5. The figure legend for figure 1 needs to be more descriptive and informative.

Minor Revisions:

Line 106: The abbreviation for transcripts per million (TMP) is provided here but the full term is not stated until line 189, TMP should be defined here first.

Lines 164-168. Should assembly be plural in these two sentences? As it reads, it seems that the authors are referring to one combined assembly of all 4 reference tissues, but given the numbers and the subsequent text this meaning does not seem to be correct and it should instead be 'assemblies'.

Lines 167-168: The use of 'tissue-specific' terminology is somewhat confusing as this denotes that the transcripts are unique to the tissue but this is clearly not the meaning here given lines 183-185 and figure 1.

Why aren't gene symbols provided for each of the genes in tables 3 and 4, if you are going to report gene symbols for some of the genes why not do so for all of the genes?

Line 248 and Line 250: Were p-values truly equal to 0 and 1 or are these rounded estimates, would p<.05 or p>.05 be more appropriate? This may be a matter of personal preference.
Very Minor/Grammatical revisions:

Line 48: The beginning of the sentence should probably read "These rodents have a distinct..."). This is one of several minor grammatical changes/typos that should be addressed but I will not belabor this as it is a very minor point.

Experimental design

Major comments:

1. Can the authors provide an explanation for the choice of male reproductive tissue for the reference tissues while leaving out the female reproductive tissue? Presumably one of the other sampled individuals was a female and tissue could have been harvested, yet only the testes were included in the reference transcriptome sequencing.

2. For the sentence from line 136-140 the authors later reference a paper for this, but the citation should probably be included here as well and addressed heteromyid rodents, not just Dipodomys.

3. Lines 138-142 Did the alignments produced contain insertions/deletions or internal stop codons? If so how were these treated for the PAML analysis? The results of the branch-sites test can be sensitive to alignment errors and with a small number of comparisons the alignments can be inspected manually to ensure this does not occur.

4. Line 143 Clustal-Omega is usually used as a multiple sequence aligner, can the authors provide details on what method it uses for producing a tree and whether branch lengths were provided to PAML or estimated in PAML?

5. This may be something planned in subsequent work but could the authors have provided kidney expression data for the 15 additional individuals or tested for differences in expression between the individuals of different sex?

Minor comments:

Line 95, do the authors mean PHRED < 2 or PHRED <20?

Line 109 should this say "using default settings"?

Validity of the findings

Major comments:

1. The authors mention calculating the site frequency spectrum for their data in line 133, did anything come of this analysis?

2. Is anything known about the demographic history of this population? As the authors acknowledge, the patterns used to infer selection according to Tajima's D can also be produced from demographic events and the authors should provide any data that exists on the population history. If this data does not exist the authors should state that this and include demography as a possible explanation of their data in their conclusions about tables 3 and 4.

3. Do the authors have data on Tajima's D for either of the genes tested in the branch sites test? If so they should report these values.

Minor comments:

Line 200: Do the authors mean complete coding sequence or open reading frame rather than 'complete exons'?

Additional comments

In this study the authors characterized the transcriptome of four separate tissues for an individual Peromyscus eremicus and conducted RNA-seq of kidney tissue of an additional 15 individuals to provide transcriptomic resources for the study of osmoregulation in this desert rodent. The study is sound in its methodology and provides a significant resource, however it does require some modifications and/or clarifications about some of the methodology and data before it could be published. In the above sections I provide suggestions and/or questions that should be addressed.

---

## Round 0.2 · Minor Revisions

Please fix the remaining typos the reviewers pointed out. Once done, I will accept your revised manuscript without further review.

·

Basic reporting

The updated methods are detailed and explicit; meets all of our criteria.

Experimental design

Good.

Validity of the findings

Good.

Additional comments

Thank you for addressing all of our concerns.

A few typos/errors remain --

p4, line 113, "assembly" => "assemble" reads
p8, line 198, "assemblies ... were accomplished" (Fix was)
p9, line 225, "are expressed _in_ all tissue types."

Reviewer 2 ·

Basic reporting

No Comments

Experimental design

No Comments

Validity of the findings

No Comments

Additional comments

The authors have sufficiently addressed the concerns raised in the previous reviews and I believe it will be ready for publication once the very minor grammatical corrections (detailed below) are addressed.

Line 113: ‘assembly’ should probably be changed to ‘assemble’

Line 118: ‘that’ should be changed to ‘than’

Line 128: ‘that’ should be changed to ‘than’

Line 189: should read “…available as part of BioProject…”

Line 198: ‘was’ should be ‘were’

Line 300: ‘These’ should be changed to ‘The’

Line 304: I think ‘test’ should be plural or ‘were’ should be changed to ‘was’

---

## Author Rebuttal · Round 0.2

# Pero Transcriptome Revision

Reviewer Comments
Reviewer 1 (C. Titus Brown)
Basic reporting
* * *
1. The authors do a good job at framing their work, showing why the study is needed, the limitations and the how the work will/can lead to future research.

2. The assembly and annotation steps were well thought out. Assemblies were error corrected, quality filtered and several steps were implemented for annotation using closely related species, Pfam database and extraction of putative coding sequences. **The only thing I wonder is why didn't the authors pool the samples when assembling.** This would not change their downstream pipeline much, however, it would help to recover low expressed transcripts. (Are there any citations for this?) **Also, I do not understand if or why the addition reads for kidney were not used for assembly.**

   *I did not conduct an assembly of the pooled samples is because I believe that tissue-specific isoforms may be reconstructed with more fidelity in this manner. I make a statement to this effect on line 111. I do trade off the potential for reconstructing additional low-coverage transcripts, but for the sake of future studies, more accurate isoform reconstruction is more important. Therefore, this is a tradeoff I am willing to take.*

   *I did not use the 15 replicate individuals in the assembly for concerns that the added polymorphism would increase run time, hardware requirements, and would decrease assembly contiguity - all effects related to a more complex de Bruijn graph. Line 114*

3. The author mentioned in results line 185 "The kidney appears to [be] an outlier in the number of unique sequences, though this could [...] result [from] the recovery of more lowly expressed transcripts [caused by] deeper sequencing." Why would this not also be the case for liver, which only has 3M (5%) less sequences?

   *I have removed this statement. I do wonder why this organ produced a larger number of contigs in the assembly, but a similar number of transcripts after filtering. I admit that I do not have a good explanation, and I hope a reader will.. The data was not of lower quality, nor are any of the assembly metrics.*

4. I am trying to understand the filtering process for the assembled reads. From my understanding (Page 4, lines 103:109) sequences were filtered using Blastn, (Page 4, lines 113:120) annotated using Blastn, HMMER3 and Transdecoder. Is my understanding correct? If so, why were the assembled sequences filtered with Blastn before annotated with Blastn and HMMER3? I thought the point of HMMER3 was to retain divergent sequences not detected by blastn.

*Yes, I filtered based on blastN to the Peromyscus maniculatus and Mus musculus transcriptomes, as well as a Mus ncRNA dataset. Given the combo of P. maniculatus being so closely related (and with a high quality annotation) and Mus being more distantly related but fantastically annotated coding regions, I am confident that I am not missing a substantial number of 'real' transcripts by employing this strategy. I think it is likely that I could have recovered more contigs using Pfam, but I would worry that the sensitivity of a HMM based search may result in the recovery of 'false' transcripts, which I am relatively intolerant of. I am electing to not filter based on Pfam, understanding the implications and potential biases of employing this strategy, though I believe them to be minimal, given that nearly all (>92%) raw reads map back to the assembly.*

5. For the natural section results, I think it would be interest to add more than two genes. Perhaps the top and bottom 10 genes from the Tajima's D analysis.

   *I have added a few more genes, Aqp 1,2,4,9. (New table 5). I did not elect to run the PAML branch site analysis on the genes identified in the analysis of Tajima's D, because for these genes, there is no a priori expectation that selection on these genes is acting differently in eremicus that in the other lineages, as there might be for genes related to osmoregulation. For the tested Aquaporins, there is no evidence of positive selection in the eremicus lineage for any of these genes, as expected. The de novo (e.g. NOT focused on candidate genes) exploration of positive selection on P. eremicus will be presented in a later work, in connection with a newly produced genome sequence.*

6. It would also be nice to have the various parts of the analysis in a repository, for reviewing and open science purposes.

   *See https://github.com/macmanes/pero_transcriptome/blob/master/analyses.md, I have also added links to portions of this document in the relevant places in the manuscript.*

> Reviewer 2
> Basic reporting

Major comments in basic reporting section:

1. Citation format should match the "name, year" format described for the journal, currently it is in a different, numbered format

   *I have fixed this*

2. Introduction, lines 46-47: In discussing that P. eremicus does not drink water, is there a study or citation that gives their lifespan and/or drinking habits? Are the authors referring back to the species account cited in the previous section?

   *I have changed the placement of the ref to make it more clear the issue of lifespan. The*

*issue of drinking, however, seems like relatively straightforward. After all, these are animals that live in the desert. Rain may happen on very rare occasions, but by definition deserts are habitats exceptionally devoid of naturally-occuring drinking water.*

3. In reporting the individuals captured the authors should provide some metadata such as age (juvenile vs. adult) and sex (were there equal numbers of each sex, or more of one sex than the other)?

   *This is done. Line 72*

4. In the methods lines 86-89, the specific multiplexing and number of lanes of sequencing should be reported (how many individuals were sequenced on each lane, etc.) Perhaps this information could be included in table 1.

   *This information is largely there. For instance, I specify that the 4 reference tissue samples were sequenced on 2 lanes of a Hiseq 2500. The 15 replicate samples were sequence across several lanes, sometimes with other samples unrelated to this project, so this number (how many individuals sequenced per lane) is not representative. Instead, the number of reads seems like the more informative number.*

5. The figure legend for figure 1 needs to be more descriptive and informative.

   *I have added more detail*

Minor Revisions:

1. Line 106: The abbreviation for transcripts per million (TMP) is provided here but the full term is not stated until line 189, TMP should be defined here first.

   *I have added the definition at the 1st use of the abbreviation, on line 125.*

2. Lines 164-168. Should assembly be plural in these two sentences? As it reads, it seems that the authors are referring to one combined assembly of all 4 reference tissues, but given the numbers and the subsequent text this meaning does not seem to be correct and it should instead be 'assemblies'.

   *Tricky grammar issue. I have changed these terms to their plural*

3. Lines 167-168: The use of 'tissue-specific' terminology is somewhat confusing as this denotes that the transcripts are unique to the tissue but this is clearly not the meaning here given lines 183-185 and figure 1.

   *Excellent point! I have modified the text. Lines 198:202. It now reads brain-derived*

4. Why aren't gene symbols provided for each of the genes in tables 3 and 4, if you are going to report gene symbols for some of the genes why not do so for all of the genes?

   *Genes were given symbols when symbols exist. For a few, e.g. h-2 class I histocompatibility antigen XM_006997718.1 has the symbol "LOC102911283". This does*

*not seem informative, so it was left off. Does this make sense to do so?*

5. Line 248 and Line 250: Were p-values truly equal to 0 and 1 or are these rounded estimates, would p<.05 or p>.05 be more appropriate? This may be a matter of personal preference.

   *These are numbers reported by PAML, so I think it is appropriate to leave them standing as is*

Very Minor/Grammatical revisions:

1. Line 48: The beginning of the sentence should probably read "These rodents have a distinct..."). This is one of several minor grammatical changes/typos that should be addressed but I will not belabor this as it is a very minor point.

   *Corrected here and a few other places*

Experimental design
Major comments:

1. Can the authors provide an explanation for the choice of male reproductive tissue for the reference tissues while leaving out the female reproductive tissue? Presumably one of the other sampled individuals was a female and tissue could have been harvested, yet only the testes were included in the reference transcriptome sequencing.

   *Thanks for bring this up. The 4 tissues were from the male animal that has been sequenced for genome assembly - these 4 tissues will aid in the annotation process. From this animal, I did not sequence all available tissues (e.g. no lung, skeletal muscle, large intestines, etc) secondary to financial constraint. Had I chosen a female animal for genome sequencing, ovary would have been included.*

2. For the sentence from line 136-140 the authors later reference a paper for this, but the citation should probably be included here as well and addressed heteromyid rodents, not just Dipodomys.

   *Excellent point. I have added the citation here and changed Dipodomys to Heteromyid to more accurately reflect the work done in that paper.*

3. Lines 138-142 Did the alignments produced contain insertions/deletions or internal stop codons? If so how were these treated for the PAML analysis? The results of the branch-sites test can be sensitive to alignment errors and with a small number of comparisons the alignments can be inspected manually to ensure this does not occur.

   *The alignments were visually inspected. While some indels do occur (especially in Homo relative to the other rodents), these all occur in units of 3nt, which to me at least suggests that they are likely accurate. The alignment column in which premature stop codons are*

*treated as missing data in PAML, though this was not common in these analyses. Line286*

4. Line 143 Clustal-Omega is usually used as a multiple sequence aligner, can the authors provide details on what method it uses for producing a tree and whether branch lengths were provided to PAML or estimated in PAML?

   *Nice catch! It is ClustalW2-Phylogeny. I have changed that in the text, added a link to that webpage, as well as indicate that branch lengths are estimated in this software. Line 168*

5. This may be something planned in subsequent work but could the authors have provided kidney expression data for the 15 additional individuals or tested for differences in expression between the individuals of different sex?

   *I would rather not, given this is outside of the current project. I actually did not even calculate individual expression values. I suppose I could if you think it would add significantly to the manuscript (I don't think it does).*

Minor comments:

1. Line 95, do the authors mean PHRED < 2 or PHRED <20?

   *Phred <2*

2. Line 109 should this say "using default settings"?

   *Fixed*

Validity of the findings
Major comments:

1. The authors mention calculating the site frequency spectrum for their data in line 133, did anything come of this analysis?

   *This SFS analysis was done as part of the calculation of Tajima's D. Nothing per se was done with the SFS itself. I will add this to Dryad, in case somebody finds it interesting or useful.*

2. Is anything known about the demographic history of this population? As the authors acknowledge, the patterns used to infer selection according to Tajima's D can also be produced from demographic events and the authors should provide any data that exists on the population history. If this data does not exist the authors should state that this and include demography as a possible explanation of their data in their conclusions about tables 3 and 4.

   *Nothing specific is known of these populations, other than what can be inferred from paleo-climate models of desert expansion over the last 10k years. I have added a statement in the interpretation of Tajima's D to reflect the possibility that demography may*

*impact these metrics. Line 257*

3. Do the authors have data on Tajima's D for either of the genes tested in the branch sites test? If so they should report these values.

   *Excellent idea. See the new table 5*

Minor comments:

1. Line 200: Do the authors mean complete coding sequence or open reading frame rather than 'complete exons'?

   *Yes, the wording has been changed to 'open reading frame'*

---

## Round 0.3 · accepted · Accept

Thanks for fixing the typos so quickly.